# Application of Lactose-Free Whey Protein to Greek Yogurts: Potential Health Benefits and Impact on Rheological Aspects and Sensory Attributes

**DOI:** 10.3390/foods11233861

**Published:** 2022-11-29

**Authors:** Agatha Transfeld da Silva, Jair José de Lima, Priscila Reis, Maurício Passos, Catiucia Giraldi Baumgartner, Aiane Benevide Sereno, Cláudia Carneiro Hecke Krüger, Lys Mary Bileski Cândido

**Affiliations:** 1Postgraduate Program in Food and Nutrition, Federal University of Paraná (UFPR), Curitiba 80210-170, Brazil; 2Department of Chemistry and Biology, Federal University of Technology—Paraná (UTFPR), Curitiba 81280-340, Brazil; 3Department of Pharmacy, Federal University of Paraná (UFPR), Curitiba 80210-170, Brazil; 4Frimesa Central Cooperative, Marechal Cândido Rondon 85960-000, Brazil

**Keywords:** β-galactosidase, rotational central composite design, physicochemical properties, sensory evaluation, microbiological analysis

## Abstract

The application of β-galactosidase in the fermentation of milk enables the acquirement of lower levels of lactose that are tolerated by lactose maldigesters and can reduce the nutritional consequences of avoiding dairy products. The present study evaluated the viability of the fortification of lactose-free prebiotic Greek yogurt formulas with whey protein concentrate (WPC). Two rotational central composite designs (RCCDs) were applied: one to perform the hydrolysis of the whey protein concentrate and another for the yogurt formulations (α = 2 with 2 central points and 4 axial points). Two β-galactosidase enzymes obtained from *Kluyveromyces lactis* were used. The content of lactose, glucose, galactose, and lactic acid were determined in the WPC, milk (pasteurized and powdered), and yogurts. The three best formulations regarding the attributes’ viscosity, syneresis, firmness, and elasticity were sensorially evaluated by using a nine-point hedonic scale. A microbiological analysis was performed after 48 h of yogurt production. The characterization of the products and the comparison of the results obtained were evaluated using the Student’s *T* test and the analysis of variance with Tukey’s test (*p*-values < 0.05). The application of a lactose-free WPC promoted viscosity, firmness, and elasticity. The syneresis was reduced, and whey increased the protein and calcium content. Lactose-free WPC can be used as a partial substitute for skimmed powdered milk in yogurts. The obtained results are encouraging with respect to the production of lactose-free Greek yogurts by the dairy industry.

## 1. Introduction

The consumption of milk and dairy products is restricted for lactase-deficient individuals [1]. Due to genetic variations occurring in some populations, lactase production is programmed to decrease over time. Asians, South Africans, and Native South Americans present the highest prevalence of reduced lactase production [2,3].

By decreasing or avoiding the consumption of milk and its derivatives, lactose-intolerant individuals can reduce their incidence of gastrointestinal symptoms [4]. Nevertheless, this strategy should be evaluated due to the potential nutritional consequences of reducing one’s protein, calcium, phosphorus, and vitamin D intake [5,6]. A cohort study with lactose-intolerant adolescents assessed that a reduction in milk consumption is associated with a reduced calcium intake [7]. Likewise, a reduction in dairy products may lead to nutritional rickets in children [8] and a low mineral density among the elderly [9].

Considering that a high intake of fermented dairy products is inversely associated with mortality and the incidence of hip fractures [10], the application of β-galactosidase in the fermentation of milk enables the acquirement of lower levels of lactose that are tolerated by lactose maldigesters [11] and is considered a growing trend in the dairy industry [12,13,14].

Greek yogurt is a fermented product with a higher total solid concentration and, consequently, a lower level of moisture than other yogurts. The preparation of this kind of product consists of the concentration of solids and the removal of acid whey to attain the desired level of solids. Alternative processes involve the fortification of milk with protein concentrates to enhance the protein content [15,16].

The addition of whey may increment the nutritional composition, mainly the protein input, and improve the attributes related to viscosity and texture [17]. Thus, whey protein concentrate can be used as a partial substitute for skimmed powdered milk [18]. The global production of whey is estimated at around 190 × 106 Ton/year; however, a large part of this production is directly disposed into drains, which makes whey a major pollutant of the dairy industry due to its high Biological Oxygen Demand (BOD), particularly due to the lactose content of this product [19,20].

The high presence of lactose in whey restricts its application in formulations designed for lactose maldigesters. Moreover, there is a growing demand from consumers for high-protein and lactose-free fermented products [12]. In this context, it is essential to find alternatives for the inclusion of lactose-free, dairy-based protein products for lactase-deficient individuals. Thus, different combinations of time and enzyme concentrations can result in a distinct index that can promote the complete lactose hydrolysis and the incorporation of whey protein concentrate into Greek-yogurts. Therefore, the aim of this study was to develop lactose-free prebiotic Greek yogurt formulas with added whey protein concentrate and calcium and that is rich in vitamin D.

## 2. Materials and Methods

### 2.1. Materials

Whey protein concentrate (WPC) was supplied by Sooro^®^ (Paraná, Brazil). The two β-galactosidase enzymes Lactozym Pure 6500 L with a ≥ 6500 NLH/g declared activity and Maxilact LGX 5000 with a ≥ 5000 NLH/g declared activity were supplied by Novozymes^®^ (Dittingen, Switzerland) and DSM Globalfood^®^ (Delft, The Netherlands), respectively. Pasteurized semi-skimmed cow’s milk used in the yogurt formula was purchased by direct delivery of Qualitat^®^ (Paraná, Brazil). Ninho Forti+ Zero lactose powdered whole milk (Nestlé^®^) and liquid vitamin D_3_ (Power Vita^®^) were purchased in local stores. The selected lactic culture was supplied by Frimesa^®^ (Paraná, Brazil); it consisted of the lactic bacteria *Streptococcus salivarius* subsp. *thermophilus* and *Lactobacillus bulgaricus* subsp. *delbrueckii* (Yomix 499 LYO 250 DCU–Danisco^®^, Paris, France). Fructooligosaccharide (FOS) was provided by Ingredion^®^ (Balsa Nova, Brazil). Doremix BL H (stabilizer/thickener), composed of gelatin (85%) and guar gum (15%), was provided by Doremus^®^ (São Paulo, Brazil).

All reagents used in this study were analytical grade. Lactose, glucose, galactose, lactic acid, and calcium carbonate standards were purchased from Sigma-Aldrich^®^ (St. Louis, MO, USA).

HPLC-grade sulfuric acid 98%, analytical grade D-(+) saccharose, D-(+) glucose, and D-(−)fructose were purchased from Sigma-Aldrich (St. Louis, MO, USA). All other chemicals and reagents were analytical or chromatographic grade.

### 2.2. Physicochemical Characterization

The pH of the samples was measured through the potentiometric method, with the use of an mPA210 model Ms Tecnopon Instrumentação^®^ pH meter (São Paulo, Brazil). Fat determination in samples was performed according to the Roese–Gottlieb 905.02 method [21]. Protein concentration was performed through the Kjeldahl method (991.20), with the use of a digester (DK20 model), recirculating water pump (JP model), purifier (SMS model), and distiller (UDK 139 model), from Velp Scientifica^®^ (Bohemia, NY, USA) [21]. Whey protein concentrate and yogurts had their calcium concentrations determined by atomic absorption spectrometry, following the 991.25 method [21]. The determination of the ash content in whey protein concentrate was performed according to the gravimetric method (930.30) [21]. The moisture of whey protein concentrate was determined according to the gravimetric method (927.05) [21].

Yogurt ash determinations were performed according to the gravimetric method (945.46) [21]. Moisture and total solid analyses of yogurts were performed according to the gravimetric method (990.20) [21]. Total dietary fiber analyses of yogurts were performed according to the enzymatic-gravimetric method (991.43) [21]. All analyses were performed in triplicate.

### 2.3. Sugars and Lactic Acid Quantification

Levels of sugars (lactose, glucose, and galactose) and lactic acid in samples of whey protein concentrate, pasteurized semi-skimmed milk, powdered milk, and yogurts were determined using high-performance liquid chromatography (HPLC).

High-performance liquid chromatographer was used, equipped with a Varian ProStar^®^ 410 model automatic injector (Victoria, Australia), ternary pump (Varian, ProStar, 230 model), and refraction index detector (Varian, ProStar, 350 model) with 40 °C detection electrochemical cell.

A column oven produced by Young Lin Instrument^®^ (Gyeonggi, Korea) was used at 55 °C with sequential use of a Phenomenex^®^ (Torrance, CA, USA) Rezek ROA column (30 cm × 7.5 cm) and a Sigma-Aldrich^®^ (St. Louis, MO, USA) Supelcogel C-610H column (30 cm × 7.8 mm). Chromatographic separations were performed with a mobile phase of sulfuric acid–water (8 mM) at a flow rate of 0.5 mL/min and injection volume of 20 µL.

The deproteinization of samples was performed according to Essig and Kleyn [22] with some adaptations. Four-gram aliquots from the samples were used, to which 4 mL of potassium ferrocyanide solution at 15% and 4 mL of zinc sulphate solution at 30% were added. After agitation and a rest for five min, the volume was raised to 50 mL with distilled water. Subsequently, samples were centrifuged at 480× *g* for 10 min. WPC sample was also filtered with a Millipore172^®^ (Darmstadt, Germany) PVDF 0.45 μm syringe filter before chromatographic analysis.

The analysis of fructose in FOS was performed with the Sigma-Aldrich^®^ (St. Louis, MO, USA) Supelcogel C-610H column (30 cm × 7.8 mm) at room temperature. The flow was 0.5 mL min, the injection volume was 20 µL, and the mobile phase was composed of ultra-pure water and 8 mM sulphuric acid.

Peaks were identified via comparison with the run time of sugars and lactic acid standards; for quantification, the external standard method was applied in triplicate, using calibration curves with five points in the concentration range between 0.01 g·100 g^−1^ and 0.45 g·100 g^−1^.

### 2.4. Experimental Planning

#### 2.4.1. WPC Hydrolysis

For the experimental planning applied to the hydrolysis of the lactose in WPC, two β-galactosidase enzymes (EC 3.2.1.23) were used. Both were obtained from *Kluyveromyces lactis*. This process was conducted at 37 °C and pH 6.

Enzymes were applied at different concentrations (0.15 to 0.30%) and times (101 to 158 min). Thus, the percentage response of hydrolysis was studied through the following independent variables: enzyme concentration (x_1_) and time (x_2_). The Rotational Central Composite Design (RCCD) was used in a 2^2^ complete factor scheme including four axis points and three central points for pure error estimation, totaling 11 tests (Table 1). This statistical design helps reduce the number of experiments and estimate all quadratic regression model coefficients and the interactional effects of the factors [23].

For each test, 10 mL of a WPC solution at 20% and variable enzyme concentrations were used. All experiments were conducted under controlled environmental conditions, using a vertical Labconco^®^ laminar air flow cabinet, class II, type B2 (Kansas City, KS, USA).

#### 2.4.2. Yogurt Formulation

For yogurt’s development, calcium carbonate (375 mg/100 mL), vitamin D_3_ (2 g/100 L), fructooligosaccharide (FOS) (3 g/100 mL), and thickener/stabilizer (0.6 g/100 mL) were added to lactose-free semi-skimmed milk. Calcium, vitamin D, and FOS concentrations were defined based on the Brazilian reference daily intake (RDI) of each substance for adults, whose values were 1000 mg, 5 μg, and ≥2.5 g, respectively. No preservatives, sweeteners, sugars, aromas, or colorings were added. The elaboration of the yogurt was based on work carried out by [11].

Pasteurized semi-skimmed milk had its lactose totally hydrolyzed by the addition of β-galactosidase at 1.5 g/L. The hydrolysis process was achieved after 120 min at 38 °C; then, we raised the temperature after this period (70 °C; 30 min) to deactivate the enzyme. This condition was defined by preliminary studies based on enzyme hydrolysis experiments [11]. The lactose-free pasteurized semi-skimmed milk was elaborated specially for this work because there was no similar product to that which was available in the market during the development of the research.

Yogurts were developed by applying different concentrations of lactose-free whey protein concentrate (LFWPC) (2.58 to 5.41%) and lactose-free powdered milk (LFPM) (2.58 to 5.41%). RCCD was employed to evaluate the effect of LFWPC or LFPM concentration on yogurt formulations. Two factors with three central points were used with a 5% significance level (Table 2). The final mix was heated at 70 °C for 30 min and placed in 145 mL plastic containers.

In sequence, lactic cultures (*Streptococcus salivarius* subsp. *thermophilus* and *Lactobacillus bulgaricus* subsp. *delbrueckii*) were inoculated. Samples were kept at 42.5 °C ± 1 °C until a pH range between 4.55 ± 0.05 was reached. Then, yogurts were cooled and kept at 5 °C ± 1 °C for 24 h to evaluate firmness and elasticity, and for 48 h to evaluate viscosity and syneresis.

Viscosity, syneresis, firmness, and elasticity responses were studied as a function of LFWPC (x_1_) and LFPM (x_2_). A sample yogurt (control) was elaborated, containing only lactose-free pasteurized semi-skimmed milk, as well as thickener/stabilizer and lactic culture in the same concentrations as the other yogurts. Finally, three treatments were selected to perform sensorial evaluation.

### 2.5. Viscosity, Syneresis, Firmness, and Elasticity

Yogurt viscosity was determined at 25 °C using a Brookfield^®^ LVDV II+ viscometer (Middleboro, MA, USA) and an adapter for small samples (S70 spiral adapter) at a 100 rpm speed. The scanning results were recorded 30 s after the beginning of the analyses.

In order to measure yogurt syneresis, the method described by Riener et al. [24] was used, with some adaptations. A total of 30 g of each sample was uniformly spread on Whatman^®^ n.1 filter paper (São Paulo, Brazil); then, these were put into funnels and placed on top of graduated cylinders. Sets were placed at 4 °C ± 1 °C for five hours. The expelled serum was collected, and the volume was recorded.

The profile analysis of yogurt texture (firmness and elasticity) was performed according to Sandoval-Castilla et al. [25], with some adaptations. Firmness and elasticity parameters were quantified as defined by Bourne [26]: firmness—the maximum required force as the test cell penetrates 30 mm into the sample; elasticity—the degree to which a sample returns to its original shape after deformation. Yogurt texture characteristics were obtained through a Brookfield^®^ CT3 Texture Analyzer (Middleboro, MA, USA). Experiments were conducted through compression tests, namely, using a cylindrical probe (TA4/1000) with 38.1 mm diameter (distance: 10 mm, test speed: 1 mm s^−1^, compression force: 4 g).

### 2.6. Microbiological Analysis

Yogurt samples were kept refrigerated (5 °C ± 1 °C) until microbiological analysis, which occurred 48 h after their production. The counting of aerobic mesophilic microorganisms, mold and yeast, total and thermotolerant coliforms, *Staphylococcus aureus*, lactic bacteria, *Salmonella*, and *Bacillus cereus* were performed following the methods recommended by the American Public Health Association [27].

Mesophile research was performed by the pour plate method in a Standard Methods Agar (PCA) at 35 °C for 48 h. Mold and yeast counting was performed by incubating the sample in a dichlorane rose-bengal chloramphenicol agar (DRBC) at 25 °C for 5 days.

For total and thermos-tolerant coliform analyses, samples were incubated at 35 °C for 48 h in lauryl sulphate tryptose broth (LST). Samples that presented gas formation were transferred to *E. coli* (EC) broth at 45 °C for 24 h and brilliant bile green broth (BGBB) at 35 °C for 48 h. In order to analyze *Staphylococcus aureus*, samples were incubated in tryptic soy broth (TSB) at 35 °C for 48 h. From each tube with growth, a culture was streaked on a Baird–Parker (BP) agar plate at 35 °C for 48 h. In case of typical colonies, they were submitted to confirmation in brain heart infusion broth (BHI) at 35 °C for 24 h and a loopful was transferred to the tryptic soy broth agar (TSB) at 35 °C for 24 h. The coagulase test enabled the identification of the MPN/g of the microorganism in question.

In order to count lactic bacteria, samples were incubated in Man, Rogosa, and Sharpe (MRS) broth at 35 °C for 4 days.

The analysis of *Salmonella* sp was performed by incubating samples in Rappaport-Vassiliadis Soya (RVS) broth at 41 °C for 24 h, and later plating them in a xylose lysine deoxycholate (XLD) agar at 37 °C for 24 h. Results were expressed with respect to absence/presence in 25 g.

In order to analyze *Bacillus cereus*, samples were forwarded to a laboratory specialized in food microbiological analysis that also used APHA standards.

### 2.7. Sensory Analysis

Yogurts were evaluated by a team of 115 non-trained assessors that received the samples in containers codified with random numbers, followed by water. The test for general yogurt acceptance was applied using a nine-point hedonic scale (Like extremely = 9 to Dislike extremely = 1). The project considered ethical aspects and was approved by the Research Ethics Committee of the Universidade Tecnológica Federal do Paraná (UTFPR) with CAAE n. 39499414.6.0000.5547. Assessors participated after agreeing to an Informed Consent Form.

The analysis was performed in individual computerized cabins, under controlled light conditions, and using the Fizz Sensory Software 2.47B program (Biosystèmes, Couternon, France).

### 2.8. Statistical Analysis

Data are presented as means ± standard deviation (SD). Data related to the characterization of the products and the comparison of the results obtained were evaluated using the SPSS Statistics program, version 22 (Chicago, IL, USA). Data distribution was verified, and the Student’s T test was performed, as well as the Analysis of Variance (ANOVA), using Tukey’s test (*p*-values < 0.05). The experimental design with the application of the Response Surface methodology was developed with the Statistics program version 7 (StatSoft) (Tulsa, OK, USA).

## 3. Results and Discussion

### 3.1. WPC Hydrolysis

Table 3 shows the WPC composition used for the experimental design and optimization of hydrolysis. The high concentration of proteins and lactose that the product naturally presents is evident, whereas the quantity of the latter disaccharide in the WPC was approximately four times higher than the average in cheese whey (5%) [28].

The analysis of the variance data, obtained from the experiment performed with the Lactozym Pure 6500 L enzyme, concluded that the effects of time and the enzyme concentration in linear and quadratic terms, as well as their interaction, were significant (*p* < 0.05). The obtained percentage variation value (R^2^) was 0.98316, indicating that 98% of the response variability may be explained by the model used. Equation (1), obtained using the coefficients resulting from the multiple regression analysis, represents the model for determining the hydrolysis percentage according to the studied variables.
Lactozym Pure 6500 L hydrolysis (%) = 83.6020 + 54.6234x1 − 73.0950x22 + 0.1421x2 − 0.0004x22 − 0.1035x1 × x2 (1)
where x1 = enzyme concentration (%); x2 = time (min)

As for the Maxilact LGX 5000 enzyme, the model may also be used for predictive purposes, since it presented a variation percentage (R^2^) equal to 0.92702. For both enzymes, the significant value of the regression model analysis also indicated the adjustment of the adopted experimental model (*p* < 0.05). Equation (2) represents the model for determining the hydrolysis percentage according to the studied variables.
Maxilact LGX 5000 hydrolysis (%) = 71.483 + 98.796x1 − 165.150x12 + 0.240x2 − 0.001x22(2)
where x1 = enzyme concentration (%); x2 = time (min)

With the goal of determining the maximum response, that is, the complete hydrolysis of the lactose in the WPC, tridimensional response surface graphs were created. Level curves help to visualize an optimal region with respect to enzyme activity, where there is a combination range of enzyme concentration and time (Figure 1).

For Lactozym Pure 6500 L, the variation in the enzyme concentration at the optimal range was from 0.25 to 0.30% and the time variation was from 125 to 150 min. For Maxilact LGX 5000, the variation in the enzyme concentration at the optimal range was from 0.23 to 0.28% and the time variation was from 125 to 150 min. These optimal ranges of enzyme concentrations and time are valuable because it is possible to assume that values within these ranges will still have the same effect in the optimized region.

The results obtained from the optimization of the WPC hydrolysis were confirmed by performing experiments under the best-defined conditions of enzyme concentration and time for each enzyme, obtaining hydrolysis degrees between 100.43 ± 0.17 (%) and 100.67 ± 0.03 (%). The hydrolysis degrees did not present statistical differences between the enzymes (*p* > 0.05).

It is important to notice that the complete hydrolysis of lactose is hardly a reachable result. This is due to factors regarding the process of enzyme hydrolysis, for example, limitations in mass transfer, with respect to immobilized enzymes, and galactose production, which has an inhibitory role towards enzyme activity [29].

Vénica et al. [30] produced, natural and sucrose-sweetened yogurts with the addition of WPC (48.2% lactose) at ratios of 1% and 2% and powdered milk (50% lactose) at ratios of 1.13% and 2.25%. After hydrolysis, using the lactase enzyme from *Kluyveromyces lactis* at concentration equal to 0.04% and a temperature from 42 °C ± 2 °C, the lactose residue obtained was 0.96 ± 0.34 (%) and 1.12 ± 0.29 (%), for natural and sweetened yogurts, respectively. No product obtained in the research could have attained the attribute “lactose;free” because their lactose concentration was higher than 100 mg of lactose per 100 g. In Denmark, Estonia, Finland, Norway, and Sweden, for example, the requirement for lactose-free products is less than 10 mg/100 g of lactose. In Germany, Slovenia, and Hungary, this threshold level is 100 mg/100 g [31].

Tests were performed to ensure the safety and feasibility at the end of each experiment. The analysis of the stable mesophilic microorganisms in the WPC from all the experiments had results lower than 36.10^3^ (CFU/g). This is lower than 1.10^5^ (CFU/g), which is the parameter required by the current law for powdered milk [32].

### 3.2. Lactose-Free Yogurt Formulation

The analysis of variance data, obtained for the yogurt viscosity, shows that the concentration effects of the lactose-free whey protein concentrate and lactose-free powdered milk in linear terms, as well as their interaction, were significant (*p* < 0.05). The obtained percentage variation value (R^2^) was 0.94714, indicating that 94.7% of the response variability may be explained by the used model (Equation (3)). In addition, the significant value of the multiple regression model analysis indicated the adjustment of the adopted experimental model (*p* < 0.05).
Viscosity (mPa·s) = 869.8396 − 82.9702x1 − 72.3534x2 + 39.2900x1x2(3)
where x1 = LFWPC concentration; x2 = LFPM concentration

The values calculated according to LFWPC and LFPM concentration for syneresis allowed us to identify that only the LFWPC variable was statistically significant at linear and quadratic levels (*p* < 0.05). The obtained value (R^2^) was 0.91144, indicating that 91% of the response variability may be explained by the used model (Equation (4)). The significant value of the multiple regression model analysis indicated the adjustment of the adopted experimental model (*p* < 0.05).
Syneresis (%) = −4.64023x1 + 0.59375x12(4)
where x1 = LFWPC concentration

The models applied according to the LFWPC and LFPM concentrations for firmness and elasticity presented R^2^ values equal to 75% and 70%, respectively, as well as a lack of model adjustment (*p* > 0.05). These models do not properly describe the relation between the independent variables and response. A change in the value range of the studied variables may adjust the models.

It is noticeable that in the higher LFWPC and LFPM concentrations, the yogurt viscosity values were higher than the ones found for the yogurts with lower LFWPC and LFPM concentrations (<4%). These higher ingredient concentrations in the yogurt formulas also enable the achievement of lower syneresis values than those that could be obtained otherwise.

The optimal range of the LFWPC and LFPM concentrations for the viscosity response was equal to 5 to 6%. For the syneresis response, the LFWPC concentrations in the optimal range remained between 4 and 5% and the LFPM concentrations between 4 and 6%.

The control yogurt sample presented lower viscosity and firmness than the WPC-enriched yogurts. Its syneresis was approximately twenty-seven times higher than that of the yogurts developed with lactose-free WPC and powdered milk. There was also an influence of the LFWPC concentrations on the elasticity attribute (Table 4).

These results indicate the beneficial action of lactose-free whey protein concentrate as compared to a standard yogurt in relation to rheological aspects. Krzeminski et al. [18] and [33] also observed the positive action of whey proteins on yogurt’s particle size, firmness, and viscosity. Whey proteins act as promoters of particle interaction and aggregation, which are important for yogurts, especially the more concentrated ones. The results of the study developed by Berber et al. [34] have shown that it is possible to completely replace non-fat dry milk in yogurt formulations with 80% WPC while enhancing textural properties such as water retention, hardness, and viscosity, and reducing syneresis. The same occurs for FOS, which is present in all the formulas at a concentration of 3%. This is regarded as a factor that increases viscosity and firmness, as verified by Seckin and Ozkilinc [35].

Therefore, due to the higher viscosity (854.00–886.00 mPas), firmness (0.1 N), and elasticity (10.27–12.12 mm), as well as the reduced syneresis (0.00–0.11%), treatments 4 (5% LFWPC/5% LFPM), 6 (5.414% LFWPC/4% LFPM), and 8 (4% LFWPC/5.414% LFPM) were selected for physicochemical characterization and sensory analysis. Table 5 shows the results of the physicochemical characteristics of these formulas. Through the chromatographic analysis, it was possible to verify the absence of lactose in these yogurt formulas (<10 mg·100 g^−1^). With the absence of lactose in the dairy product and the consequent increase in monosaccharides resulting from hydrolysis, the lactic bacteria used mainly glucose as the main substrate to provide energy and release lactic acid; thus, the galactose concentrations were higher than glucose concentrations.

The Food and Agriculture Organization (FAO) and World Health Organization (WHO) require yogurt acidity (lactic acid g/100 g) to be at least 0.6 [36]. These three formulations (4, 6, and 8) comply with such lactic acid requirements.

In addition, the protein concentration in yogurt remained between 7.41 and 8.04 g/100 g. The yogurts produced had a protein concentration above the average of lactose-free Greek yogurts with no addition of fruit pulps or other preparations. This high protein concentration is explained by the addition of whey protein concentrate. The addition of whey protein concentrate appears to be a feasible alternative for the dairy industry, not only because it supports an increased protein intake, but also because it improves some sensory characteristics of interest, such as texture and viscosity. In addition to the quantitative perspective, it is important to consider the protein quality, since whey proteins present all essential amino acids, including branched chain amino acids (leucine, isoleucine, and valine), and are recognized as proteins with a high biological value and that can be quickly absorbed [34,37,38]

As for fat, formula 4 presented a higher concentration than the others, but there was no significant statistical difference. Tamime and Robinson [39] researched the fat concentration of different Greek yogurts produced by ultrafiltration, fat addition, or by mechanical separators; the average concentration was from 6.1 to 9.2 (g/100 g).

In this study, the method used to obtain Greek yogurts was the addition of ingredients (a stabilizer/thickener, whey protein concentrate, powdered milk, and fructooligosaccharide) that promote characteristics of interest in Greek yogurt, and that are nutritionally interesting for lactose-intolerant people.

The ash concentration in this study’s yogurts (1.37 ± 0.06 to 1.39 ± 0.06 g/100 g) is high when compared to other concentrated yogurts. Serafeimidou et al. [40] researched twenty-four samples of Greek yogurts sold in Greece, and the ash concentrations remained between 0.632 ± 0.022 to 1.107 ± 0.006 (g/100 g).

The increased content of calcium in our yogurts was due to the addition of calcium carbonate. When compared to other sources commonly used in food, calcium carbonate has the highest concentration of the mineral [41]. Through calcium supplementation, it was possible to reach 15.56% to 22% of the daily recommendation for adults (1000 mg day) [42]. The yogurts with higher LFWPC concentrations (≥5%) obtained higher calcium levels, since the LFWPC contained 806.21 ± 5.40 (mg/100 g) calcium, whereas for LFPM the content was equal to 590.19 ± 1.11 (mg/100 g).

The products also reached 40% of the recommended daily intake of vitamin D for adults (5 µg day) [42].

Through the addition of fructooligosaccharide (FOS), the three yogurt formulas could be labeled as having prebiotic properties. This product was analyzed by HPLC, whereby the presence of 8% fructose in the free form was verified, indicating that the product was intact and would act as dietary fiber.

Fructooligosaccharide is a non-digestible substance that acts as a prebiotic and modulates intestinal microbiota, thus having an important function for lactose-intolerant people, who often present an imbalance in the composition of endogenous bacteria in their intestines. Fructooligosaccharide is fermented by the beneficial bacteria inside the colon, lactobacilli and bifidobacteria, forming short-chain fatty acids that can boost enterocyte cell proliferation, stimulate the function of the immune system, and regulate fat metabolism, thereby limiting lipogenesis and cholesterol synthesis [43]. This component has a beneficial effect on mineral absorption, mainly with respect to calcium. The production of short-chain fatty acids promotes the decrease in the intestinal pH, which turns insoluble calcium into its ionic form, helping its absorption [44]. In addition to nutritional benefits, fructooligosaccharide (FOS) presents technological properties, being a substitute for sucrose and possessing a great water retention capacity, high solubility and stability, and offering an excellent amount of energy [45].

### 3.3. Microbiological Analysis

The average count of lactic bacteria was equal to 6.90 ± 0.63 log CFU/g for sample 4, 6.88 ± 0.67 log CFU/g for sample 6, and 6.91 ± 0.68 log CFU/g for sample 8. These results respect the minimum lactic bacteria count established by the FAO and WHO for yogurts (10^7^ CFU/g). The results of the microbiological analyses of the three yogurt formulas (4, 6 and 8) showed lower counts than the upper limits established by the local legislation [32,46]]. The yogurts presented a count of mesophiles < 4.73.10^3^ CFU/g, total coliforms < 9Bra3 MPN/g and thermotolerants < 3 MPN/g, a count of *Staphylococcus aureus* < 23 MPN/g, no molds and yeasts, and a *Bacillus* cereus count < 1.10 CFU/g. There were no *Salmonella* sp in the three samples analyzed.

### 3.4. Sensory Evaluation

The general acceptability averages of formulas 4, 6, and 8 were: 6.61 ± 1.71, 6.7 ± 1.74, and 6.57 ± 1.78, respectively. The formula with the highest concentration of whey protein concentrate (5.41%) resulted in a higher general acceptance than the formulas with lower concentrations of this ingredient. However, there was no statistical difference among the three tested formulas (*p* > 0.05).

Although it is a common practice to add sugar to improve the sensory characteristics of food products, no sugars or sweeteners were added to these yogurts. Even without the addition of sugar and artificial sweeteners, one noticeable characteristic in the yogurts was their sweetness, which is attributable to the fact that the sugars resulting from the hydrolysis of lactose have a stronger sweetening power than lactose itself [47].

## 4. Conclusions

The main highlights of this research are related to the process of obtaining a lactose-free product by applying enzymes and time under optimized conditions. The best hydrolysis condition, defined by using the response surface methodology, had a variation in the enzyme concentration from 0.23 to 0.30% and a time variation that was from 125 to 150 min. The optimization of hydrolysis promoted the absence of lactose in the yogurt formulas (<10 mg·100 g^−1^).

The obtained results revealed that the addition of lactose-free whey protein concentrate in Greek yogurts promoted their rheological characteristics and sensory properties. The yogurt formulas with 4–5.41% lactose-free WPC presented higher viscosity (854.00–886.00 mPas), firmness (0.1 N), and elasticity (10.27–12.12 mm), as well as reduced syneresis (0.00–0.11%). Moreover, adding this ingredient contributed to the increase in the protein (7.41 and 8.04 g/100 g) and calcium content, which were allowed to reach 15.56% to 22% of the daily recommendation for adults. The yogurt formulas were well accepted by the assessors with a general acceptance of 6.57 ± 1.78 to 6.7 ± 1.74. The results demonstrate that the addition of lactose-free whey protein concentrate in Greek yogurts promotes both their rheological characteristics and sensory properties.

## Figures and Tables

**Figure 1 foods-11-03861-f001:**
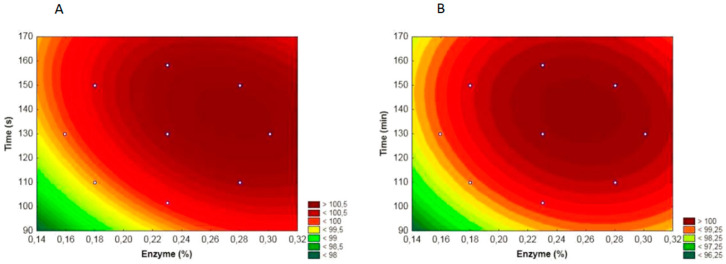
Effect of β-galactosidase enzymes on lactose hydrolysis of whey protein concentrate as a function of enzyme concentration (%) and time (min). (**A**) Lactozym Pure 6500 L; (**B**) Maxilact LGX 5000.

**Table 1 foods-11-03861-t001:** Codified variables with variation levels for the hydrolysis of whey protein concentrate.

Factors	Code	−1.41	−1	0	1	1.41
**Lactose hydrolysis**						
Enzyme (%)	x_1_	0.159	0.18	0.23	0.28	0.30
Time (min)	x_2_	101.7	110	130	150	158.28

**Table 2 foods-11-03861-t002:** Codified variables with variation levels for yogurt formulas.

Factors	Code	−1.41	−1	0	1	1.41
**Yogurt formulation**						
LFWPC (%)	x_1_	2.585	3	4	5	5.414
LFPM (%)	x_2_	2.585	3	4	5	5.414

LFWPC: lactose-free whey protein concentrate; LFPM: lactose-free powdered milk.

**Table 3 foods-11-03861-t003:** Initial composition of the whey protein concentrate.

	Results
Lactose (g 100 g^−1^)	22.96 ± 0.07
Glucose (g 100 g^−1^)	0.62 ± 0.01
Galactose (g 100 g^−1^)	0.69 ± 0.02
pH	6.05 ± 0.01
Acidity (g lactic acid 100 g^−1^)	0.67 ± 0.00
Fat (g 100 g^−1^)	5.50 ± 0.00
Protein (g 100 g^−1^)	60.07 ± 0.08
Moisture (%)	5.64 ± 0.40
Ashes (%)	3.94 ± 0.03

Average ± average standard deviation, *n* = 3.

**Table 4 foods-11-03861-t004:** Values of viscosity, firmness, elasticity, and syneresis of the formulas.

Formulas	Viscosity (mPa s)	Firmness (N)	Elasticity (mm)	Syneresis (%)
Control sample	352.00 ± 2.64 ^i^	0.03 ± 0.00 ^g^	8.36 ± 0.22 ^ghij^	54.91 ± 1.69 ^a^
1	726.16 ± 2.46 ^e^	0.07 ± 0.00 ^f^	9.00 ± 0.03 ^efj^	1.75 ± 0.00 ^b^
2	739.00 ± 1.00 ^d^	0.07 ± 0.00 ^f^	9.27 ± 0.12 ^cdefh^	1.39 ± 0.6 ^b^
3	716.00 ± 1.73 ^f^	0.08 ± 0.00 ^de^	9.29 ± 0.37 ^cdefg^	1.16 ± 1.00 ^b^
4	886.00 ± 3.60 ^a^	0.1 ± 0.00 ^ab^	10.43 ± 0.07 ^b^	0.11 ± 0.19 ^b^
5	674.66 ± 4.16 ^g^	0.09 ± 0.00 ^ce^	9.24 ± 0.38 ^cdef^	2.07 ± 0.52 ^b^
6	854.66 ± 3.51 ^b^	0.1 ± 0.00 ^a^	12.12 ± 0.83 ^a^	0.11 ± 0.20 ^b^
7	666.00 ± 2.64 ^h^	0.07 ± 0.00 ^f^	9.57 ± 0.33 ^bf^	0.57 ± 0.99 ^b^
8	862.33 ± 2.30 ^b^	0.1 ± 0.00 ^a^	10.27 ± 0.49 ^bc^	0.00 ± 0.00 ^b^
9	776.00 ± 1.00 ^c^	0.09 ± 0.00 ^cd^	9.88 ± 0.16 ^be^	0.11 ± 0.20 ^b^
10	768.33 ± 3.51 ^c^	0.09 ± 0.00 ^bc^	10.17 ± 0.45 ^bd^	0.11 ± 0.20 ^b^

Average ± average standard deviation, *n* = 3. Averages in the same column with different superscript letters have statistical differences (*p* < 0.05).

**Table 5 foods-11-03861-t005:** Physical-chemical characterization of formulas.

	Formula 4	Formula 6	Formula 8
Lactose (g 100 g^−1^)	0.0	0.0	0.0
Glucose (g 100 g^−1^)	1.47 ± 0.08 ^a^	1.25 ± 0.18 ^a^	1.26 ± 0.05 ^a^
Galactose (g 100 g^−1^)	1.65 ± 0.31 ^a^	1.56 ± 0.28 ^a^	1.71 ± 0.07 ^a^
Lactic Acid (g 100 g^−1^)	1.30 ± 0.01 ^a^	1.49 ± 0.03 ^a^	1.46 ± 0.09 ^a^
pH	4.81 ± 0.01 ^a^	4.79 ± 0.00 ^a^	4.79 ± 0.00 ^a^
Protein (g 100 g^−1^)	8.04 ± 0.70 ^a^	7.94 ± 0.39 ^a^	7.41 ± 0.40 ^a^
Fat (g 100 g^−1^)	6.05 ± 1.01 ^a^	5.26 ± 0.83 ^a^	4.51 ± 0.87 ^a^
Ashes (g 100 g^−1^)	1.39 ± 0.06 ^a^	1.38 ± 0.06 ^a^	1.37 ± 0.16 ^a^
Calcium (mg 100 g^−1^)	215.16 ± 1.94 ^a^	220.05 ± 5.42 ^a^	155.68 ± 5.27 ^b^
Total fibers (g 100 g^−1^)	3.05 ± 0.05 ^a^	3.02 ± 0.04 ^a^	3.00 ± 0.03 ^a^
Moisture (%)	79.35 ± 0.37 ^a^	79.25 ± 0.28 ^a^	79.29 ± 0.29 ^a^
Total solids (g 100 g^−1^)	20.86 ± 0.37 ^a^	20.74 ± 0.28 ^a^	20.70 ± 0.29 ^a^

Average ± average standard deviation, *n* = 3. Averages in the same line with different superscript letters have statistical differences (*p* < 0.05).

## Data Availability

Data can be made available upon reasonable request from the corresponding author.

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
