# Peer review of "Application of Lactose-Free Whey Protein to Greek Yogurts: Potential Health Benefits and Impact on Rheological Aspects and Sensory Attributes"

_foods, 2022, doi:10.3390/foods11233861_

Round 1

Reviewer 1 Report

The manuscript addresses an important aspect of the production of lactose-free fermented dairy products as a response to increasing consumer demand for lactose-free products. The topic is interesting, however, the manuscript needs some revisions:

1.The authors should demonstrate the novelty of their study to a greater degree.

2. Duplicated part in materials and methods section - 64-75 and 79-89

3. Line 52 - please use superscript 106 ton

4. What is the difference between both enzymes used?

5. Line 134 - please specify the values

6. Line 231 - CFu/g-1 is used whereas in line 306 log CFU, please unify

7. Particles size measurement  would be an interesting addition

8. Lines 310-313 why some values are in MPN?

Author Response

The manuscript addresses an important aspect of the production of lactose-free fermented dairy products as a response to increasing consumer demand for lactose-free products. The topic is interesting, however, the manuscript needs some revisions:

1.The authors should demonstrate the novelty of their study to a greater degree.

The innovation of the study was highlighted.

  1. Duplicated part in materials and methods section - 64-75 and 79-89

 Duplicated data was removed from the article.

  1. Line 52 - please use superscript 106 ton

 Ton was replaced by TON

  1. What is the difference between both enzymes used?

 The difference between both enzymes was stated.

  1. Line 134 - please specify the values

The values reference daily intake values (RDI) were specified.

  1. Line 231 - CFu/g-1 is used whereas in line 306 log CFU, please unify

In order to perform statistical analysis of the data, it is convenient to convert  potencies of 10 into logarithms. This way you get smaller numbers, which reduces the error and the coefficient of variation. In this case, for example, when we have 6.90 ±0.63 log, to know in potency of 10 how much this value is, we must do the inverse operation. that is, 10 to the potency of 6.00 which is equal to 7.94x106

  1. Particles size measurement  would be an interesting addition

Thank you for the suggestion, however this technique was not among the objectives of the research when the work was planned and the yogurt samples were prepared. 

  1. Lines 310-313 why some values are in MPN?

The microbiological analysis were performed according to the methods recommended by

American Public Health Association (APHA). Thus, the analysis results were expressed accordingly.

Reviewer 2 Report

I reviewed the manuscript entitled, Application of lactose-free whey protein on Greek yogurts: potential health benefits and impact on rheological aspects and sensory attributes. The manuscript is well written and has scientific soundness. Introduction, methodology, research and discussion are appropriate and contribute to the field. In my opinion, this manuscript need  address below suggestions.

Line 76: degree can be replaced with grade

Section 2.3. HPLC conditions and other details must be added

Section 2.4.2. reference should be added for yogurt formulations

Line 156: add the Table title

In order to measure yogurt syneresis, the method described by Riener et al. [23] was 164 used, with adaptations… provide the details of methodology

Provide the details of texture (firmness and elasticity) methodology

2.6. Microbiological analysis: provide the details of methodology

2.7. tasters can be replaced with most appropriate word

Line 247: What is overwritten? Is it superscripts?

Why are authors selected formulations 4, 6, and 8?

3.3 and 3.4: discussion must be improved

Conclusions: please revise the conclusions to reflect the findings. Also, include future recommendations

References are not according to the journal format. Please revise it

Author Response

I reviewed the manuscript entitled, Application of lactose-free whey protein on Greek yogurts: potential health benefits and impact on rheological aspects and sensory attributes. The manuscript is well written and has scientific soundness. Introduction, methodology, research and discussion are appropriate and contribute to the field. In my opinion, this manuscript need  address below suggestions.

Line 76: degree can be replaced with grade

 The word degree was replaced by grade.

Section 2.3. HPLC conditions and other details must be added

HPLC conditions were described in detail.

Section 2.4.2. reference should be added for yogurt formulations

Reference was added

Line 156: add the Table title

 The table title was added.

In order to measure yogurt syneresis, the method described by Riener et al. [23] was 164 used, with adaptations… provide the details of methodology

 The methodology used by Riener et al. [23] was fully described.

Provide the details of texture (firmness and elasticity) methodology

The details were provided

2.6. Microbiological analysis: provide the details of methodology

 The details of all microbiological analysis performed were fully described.

2.7. tasters can be replaced with most appropriate word

 The term taster was replaced by assessor.

Line 247: What is overwritten? Is it superscripts?

The word overwritten was replaced by superscript.

Why are authors selected formulations 4, 6, and 8?

The information was inserted in the main text: Due to higher viscosity (854.00-886.00 mPas), firmness (0.1 N) and elasticity (10.27-12.12 mm), as well as reduced syneresis (0.00-0.11 %), the treatments 4 (5% LFWPC/5% LFPM), 6 (5.414% LFWPC/4% LFPM) and 8 (4% LFWPC/5.414% LFPM) were selected for physicochemical characterization and sensory analysis.

3.3 and 3.4: discussion must be improved

the discussion was improved

Conclusions: please revise the conclusions to reflect the findings. Also, include future recommendations

The  conclusion was revised

References are not according to the journal format. Please revise it

References were revised and prepared using Mendeley

Reviewer 3 Report

This article " Application of lactose-free whey protein on Greek yogurts: potential health benefits and impact on rheological aspects and sensory attributes” was revised and have a novelty and I recommend consideration of the following comments.

Title: If you can rewrite and make it more interesting for readers. I propose: “Application of lactose-free whey protein concentrate on Greek yogurts: potential health benefits and impact on rheological aspects and sensory attributes”.

Abstract:

·         The type of statistical design used in this research should be mentioned including the number of central, axial, and factorial points etc.

·         The abstract is very concise and not comprehensive. Please provide it complete and perfect.

·         The statistical comparing was not done.

·         The sensory evaluation and microbial analysis were not done.

Keywords: Please choose keywords other than the main words of the title. In this case, other researchers can find your article by searching a wide range of words through databases. I propose another keywords as the follow:

whey protein concentrate; lactose; lactose-free; enzyme hydrolysis; yoghurt, β-galactosidase enzyme, Rotational central composite design, Physicochemical properties, Sensory evaluation, Microbiological analysis

Abbreviation:

·         Please provide “Abbreviation section consequent the Keywords

Introduction:

·         Please explain to response surface analysis in one paragraph and you can help from the two address:

·         Journal of Food Processing and Preservation 45 (4), e15311

·         https://doi.org/10.1590/fst.52120

·         Line 59-60: Please explain as more complete the treatments and statistical design as detail and comprehensive.

Materials and Methods:

·         Line 93-102: The way of expressing the method of measuring macronutrients and other parameters has a scientific flaw. Please cite and take help from the following article for the correct way of expressing it, so that the standard number of the working method should be clearly stated (https://doi.org/10.1590/fst.60820).

·         Line 133: FOS? Expression is very confused.

·         Line 138: why 1.5 g/L of enzyme was utilized?

·         Please add the combination of independent and dependent variables for the both of your response surface methology as two separate tables to the text of the article so that the readers have a better understanding. At the same time, the regression equations of modeling should also be done, for a better understanding, I draw your attention to the following articles.

·         Journal of Food Processing and Preservation 45 (5), e15456

·         Journal of Food Measurement and Characterization 15 (1), 495-507

·         Journal of Food Processing and Preservation 44 (8), e14563

·         Journal of Food Measurement and Characterization 14 (6), 3216-3226

 “Results:

·         In the throughout the text of the article, the average number is sufficient and there is no need to state the standard deviation or standard error.

·         Table 3: why did you compare only the formula 4, 6, and 8?

·         From the point of view of the statistical design, the tables and figures related to Response Surface methodology are very weak. I hope that with the help of the suggested references, he can improve the statistical part of his text.

Discussion:

·         Discussion text must grammar improve and in some cases it is very weak and maybe there is no discussion at all.

Conclusions:

·         Conclusion is very general, try to make it more scientific, comprehensive and concise in detail, especially.

·         You ought to conclude only study results not to expressing the general sentences.

References:

·         The references are too old please include the new and current ones.

The article has many flaws in express and concept of English, it is suggested to be revised in a scientific and native way.

Author Response

This article " Application of lactose-free whey protein on Greek yogurts: potential health benefits and impact on rheological aspects and sensory attributes” was revised and have a novelty and I recommend consideration of the following comments.

Title: If you can rewrite and make it more interesting for readers. I propose: “Application of lactose-free whey protein concentrate on Greek yogurts: potential health benefits and impact on rheological aspects and sensory attributes”.

Abstract:

  •         The type of statistical design used in this research should be mentioned including the number of central, axial, and factorial points etc.
  •         The abstract is very concise and not comprehensive. Please provide it complete and perfect.
  •         The statistical comparing was not done.
  •         The sensory evaluation and microbial analysis were not done.

The information was added

Keywords: Please choose keywords other than the main words of the title. In this case, other researchers can find your article by searching a wide range of words through databases. I propose another keywords as the follow:

whey protein concentrate; lactose; lactose-free; enzyme hydrolysis; yoghurt, β-galactosidase enzyme, Rotational central composite design, Physicochemical properties, Sensory evaluation, Microbiological analysis

Keywords (whey protein concentrate; lactose; lactose-free; enzyme hydrolysis; yoghurt,) were replaced by ( β-galactosidase, rotational central composite design, physicochemical properties, sensory evaluation, microbiological analysis)

Abbreviation:

  •         Please provide “Abbreviation section consequent the Keywords

We chose to use the manuscript preparation in which it is suggested that “Abbreviations should be defined the first time they appear in each of three sections: the abstract; the main text; the first figure or table. When defined for the first time, the acronym/abbreviation/initialism should be added in parentheses after the written-out form.” so we have revised all the abbreviations in the text to ensure that they  are defined in the main text

Introduction:

  •         Please explain to response surface analysis in one paragraph and you can help from the two address:

Information was added

  •         Journal of Food Processing and Preservation 45 (4), e15311
  •         https://doi.org/10.1590/fst.52120
  •         Line 59-60: Please explain as more complete the treatments and statistical design as detail and comprehensive.

Materials and Methods:

  •         Line 93-102: The way of expressing the method of measuring macronutrients and other parameters has a scientific flaw. Please cite and take help from the following article for the correct way of expressing it, so that the standard number of the working method should be clearly stated (https://doi.org/10.1590/fst.60820).

the way of expressing was corrected

  •         Line 133: FOS? Expression is very confused.

corrected for fructooligosaccharide (FOS)

  •         Line 138: why 1.5 g/L of enzyme was utilized?

Information was added. This condition was defined by preliminary studies of our group, based on the enzyme hydrolysis experiments

  •         Please add the combination of independent and dependent variables for the both of your response surface methology as two separate tables to the text of the article so that the readers have a better understanding. At the same time, the regression equations of modeling should also be done, for a better understanding, I draw your attention to the following articles.

We have separated the design into specific tables for hydrolysis and for yogurt formulation

  •         Journal of Food Processing and Preservation 45 (5), e15456
  •         Journal of Food Measurement and Characterization 15 (1), 495-507
  •         Journal of Food Processing and Preservation 44 (8), e14563
  •         Journal of Food Measurement and Characterization 14 (6), 3216-3226

 “Results:

  •         In the throughout the text of the article, the average number is sufficient and there is no need to state the standard deviation or standard error.

Standard deviation for WPC analysis was included

  •         Table 3: why did you compare only the formula 4, 6, and 8?

The information was inserted in the main text: Due to higher viscosity (854.00-886.00 mPas), firmness (0.1 N) and elasticity (10.27-12.12 mm), as well as reduced syneresis (0.00-0.11 %), the treatments 4 (5% LFWPC/5% LFPM), 6 (5.414% LFWPC/4% LFPM) and 8 (4% LFWPC/5.414% LFPM) were selected for physicochemical characterization and sensory analysis.

  •         From the point of view of the statistical design, the tables and figures related to Response Surface methodology are very weak. I hope that with the help of the suggested references, he can improve the statistical part of his text.

the statistical part was improved

Discussion:

  •         Discussion text must grammar improve and in some cases it is very weak and maybe there is no discussion at all.

the discussion was improved

Conclusions:

  •         Conclusion is very general, try to make it more scientific, comprehensive and concise in detail, especially.
  •         You ought to conclude only study results not to expressing the general sentences.

The  conclusion was revised

References:

  •         The references are too old please include the new and current ones.

New references were included

The article has many flaws in express and concept of English, it is suggested to be revised in a scientific and native way.

English was revised

Round 2

Reviewer 2 Report

Authors are now answered the suggestions made by me. In my opinion, this version of the manuscript can be accepted for publication in Foods.

Reviewer 3 Report

The manuscript was revised according to the suggestions and it is acceptable in this format.